# The Genotype-Phenotype Correlation in Human 5α-Reductase Type 2 Deficiency: Classified and Analyzed from a SRD5A2 Structural Perspective

**DOI:** 10.3390/ijms24043297

**Published:** 2023-02-07

**Authors:** Jieun Seo, Saeam Shin, Sang-woon Kim, Su Jin Kim, Myeongseob Lee, Kyungchul Song, Junghwan Suh, Seung-Tae Lee, Yong Seung Lee, Hyun Wook Chae, Ho-Seong Kim, Jong Rak Choi, Sangwon Han, Ahreum Kwon

**Affiliations:** 1Department of Laboratory Medicine, Severance Hospital, Yonsei University College of Medicine, Seoul 03722, Republic of Korea; 2Department of Urology, Urological Science Institute, Yonsei University College of Medicine, Seoul 03722, Republic of Korea; 3Department of Pediatrics, Severance Children’s Hospital, Institute of Endocrinology, Yonsei University College of Medicine, Seoul 03722, Republic of Korea

**Keywords:** 5α-reductase deficiency, *SRD5A2* gene, disorders of sex development, genotype-phenotype correlation, external masculinization score

## Abstract

The phenotype of the 5α-reductase type 2 deficiency (5αRD2) by the *SRD5A2* gene mutation varies, and although there have been many attempts, the genotype-phenotype correlation still has not yet been adequately evaluated. Recently, the crystal structure of the 5α-reductase type 2 isozyme (SRD5A2) has been determined. Therefore, the present study retrospectively evaluated the genotype-phenotype correlation from a structural perspective in 19 Korean patients with 5αRD2. Additionally, variants were classified according to structural categories, and phenotypic severity was compared with previously published data. The p.R227Q variant, which belongs to the NADPH-binding residue mutation category, exhibited a more masculine phenotype (higher external masculinization score) than other variants. Furthermore, compound heterozygous mutations with p.R227Q mitigated phenotypic severity. Similarly, other mutations in this category showed mild to moderate phenotypes. Conversely, the variants categorized as structure-destabilizing and small to bulky residue mutations showed moderate to severe phenotypes, and those categorized as catalytic site and helix-breaking mutations exhibited severe phenotypes. Therefore, the SRD5A2 structural approach suggested that a genotype-phenotype correlation does exist in 5αRD2. Furthermore, the categorization of *SRD5A2* gene variants according to the SRD5A2 structure facilitates the prediction of the severity of 5αRD2 and the management and genetic counseling of patients affected by it.

## 1. Introduction

The 5α-reductase type 2 deficiency (5αRD2, OMIM #264600) is one of the disorders/differences of sex development (DSD). It is an autosomal recessive disease caused by the *SRD5A2* gene mutation (OMIM #607306) [1]. The *SRD5A2* gene encodes 5α-reductase type 2 isozyme (SRD5A2), which catalyzes the conversion of testosterone (T) to dihydrotestosterone (DHT) using nicotinamide-adenine dinucleotide phosphate (NADPH) as a cofactor [2,3,4]. Defects in the *SRD5A2* gene, and the resultant inability to convert T to DHT [1], elicit an increased T/DHT ratio at baseline or after human chorionic gonadotropin (hCG) stimulation [2]. Although T influences the development of the Wolffian ducts, DHT has 10-fold higher potency than T and is essential for the development of the normal male external genitalia, prostate, and urethra [5]. Therefore, patients with 5αRD2 present a wide range of genital ambiguity, from mild undervirilized male to complete female external genitalia. Moreover, the severity of the phenotype varies depending on the degree of retained SRD5A2 activity [6,7,8].

As highly variable manifestations make it difficult to distinguish 5αRD2 from other DSDs, genetic analysis of the *SRD5A2* gene is essential for the diagnosis of 5αRD2. To date, the Human Gene Mutation Database (HGMD) [9] has reported approximately 186 *SRD5A2* gene mutations in the coding region of the gene, including 121 missense and 10 nonsense mutations, 12 splice site mutations, 1 localized in the regulatory region, 20 small deletions, 9 small insertions, 5 small indels, and 6 large deletions. Notably, the majority of these are missense mutations located throughout the whole gene.

There have been many attempts to elucidate a genotype-phenotype link in 5αRD2 [8,10,11,12,13], as the wide range of phenotypes is likely attributable to individual genetic backgrounds. However, most studies have not established a genotype-phenotype correlation [7,12,14,15,16]. Moreover, phenotypic variability has been reported even in individuals with the same *SRD5A2* mutation [12,14,17] and in siblings [8,14,18,19]. Therefore, until recently, no genotype-phenotype correlation was thought to exist in 5αRD2 [12].

Recently, the molecular mechanisms of the SRD5A2 catalysis reaction were unveiled by structural analysis [20]. The evaluation of the genotype-phenotype correlation may necessitate approaching the mutations according to the molecular mechanisms and structure of 5αRD2 instead of focusing on each mutation. From a structural perspective, there are five categories of *SRD5A2* gene mutations: (1) catalytic site mutations, (2) NADPH-binding residue mutations, (3) structure-destabilizing mutations, (4) helix-breaking mutations, and (5) small to bulky residue (destabilizing) mutations [21]. For example, the p.R227Q variant of the *SRD5A2* gene, one of the founder mutations in East Asians [7,8,13,22], reduces the activity of SRD5A2 by disrupting cofactor (NADPH) binding [20,21]. Furthermore, the p.H231R variant, which is widely distributed among Caucasians [10], is also associated with the NADPH binding residue [21]. In contrast, the p.R246Q variant, which has been frequently reported in various ethnic origins [10], has been suggested to be associated with structural destabilization [20,21]. Similarly, the p.Q126R, p.P181L, and p.P212R variants also belong to this category [21]. The p.G203S variant, which is another frequently reported mutation in East Asian patients [8,13,19,23,24], belongs to the small to bulky residue category along with p.G34R and p.G196S [21]. However, an evaluation of the genotype-phenotype correlation from a *SRD5A2* gene structural perspective is yet to be demonstrated.

The aim of this study was to determine the genotype-phenotype correlation in 19 Korean patients with genetically confirmed 5αRD2. This was accomplished by comparing clinical findings between patients with or without p.R227Q and p.R246Q variants, which were the most prevalent in this cohort. In addition, the correlation between genotype and phenotype was evaluated in relation to the structure of *SRD5A2*. This was achieved by classifying data from previous studies reported in the HGMD database according to five crystal-structure categories of SRD5A2 [21], and then comparing the severities of the phenotypes. 

## 2. Results

### 2.1. Patients and SRD5A2 Gene Analysis

Out of 143 patients with 46,XY DSD, the data of 19 patients with 5αRD2 were analyzed. The median age at which the patients first visited the hospital for evaluation was 0.6 years (range: 7 days–16.4 years), and the median age at which molecular analysis was performed was 7.9 years (range: 2 months–23.7 years). There was neither consanguinity history nor family history of 5αRD2.

In the 19 patients, we identified 8 different mutations: 6 (31.6%) patients presented with homozygous mutations, whereas the remainder presented with compound heterozygous mutations (Table 1). Of these 8 different mutations, p.T53R has not been previously described. 

All the mutant alleles were found to be located in exons 1 (7, 18.4%), 4 (13, 34.2%), and 5 (18, 47.4%). The most common mutant alleles were p.R246Q (17 alleles, 44.7%), followed by p.R227Q (10 alleles, 26.3%), p.Q6* (5 alleles, 13.2%), and p.G203S (2 alleles, 5.3%) (Appendix A). Among the 19 patients, 13 (34.2%) had the p.R246Q mutant allele, and 9 (23.7%) had the p.R227Q allele.

### 2.2. Phenotype and the Correlation with Genotype

Nineteen patients were classified according to their molecular results and the main clinical features noted during their first visit to our clinic, as summarized in Table 1. Half of the patients presented with micropenis with various degrees of hypospadias (9 of 19, 23.7%), whereas the other half presented with either female external genitalia with clitoromegaly (5 of 19, 13.2%) or normal female genitalia (4 of 19, 10.5%). Only 3 patients presented with isolated micropenis (15.8%). Although the external masculinization score (EMS) varied from 2.0–9.0, the median EMS was 3.0, indicating a predominantly female phenotype (Table 1). 

Of note, the median EMS of patients with the p.R227Q mutation was significantly higher than that of patients without this mutation (*p* < 0.001), indicating that the p.R227Q variant was less severe and elicited a phenotype closer to male than others (Figure 1). Moreover, all of the patients with isolated micropenis had the p.R227Q variant allele. The incidences of cryptorchidism were significantly lower in the p.R227Q variant group as opposed to the non-p.R227Q group (Rt.: *p* = 0.015 and Lt.: *p* = 0.004). The incidence of hypospadias was also lower in the p.R227Q variant group as opposed to the non-p.R227Q group; however, this difference was not statistically significant. 

In contrast, patients with the p.R246Q variant presented a more severe phenotype than others. Although not statistically significant, the median EMS was lower and the frequency of cryptorchidism was higher in the p.R246Q variant group than in the non-p.R246Q variant group (Figure 2). 

Of the 19 participants, 6 (31.6%) were initially oriented toward the female sex and had a median EMS of 2.0 (range: 2.0–2.5). Five of these patients were diagnosed with 5αRD2 while being raised female, whereas the remaining patient was evaluated for ambiguous genitalia at 1 month of age. It was decided to raise the patient as female in consideration of the symptoms. A switch from female to male identity was requested by two patients (patients 14 and 15) in the peripubertal and postpubertal periods, respectively. All of the patients in the p.R227Q variant group were assigned to the male sex; therefore, female sex assignment in the non-p.R227Q variant group (6, 60%) was higher (*p* = 0.008). In contrast, all of the female-assigned participants had the p.R246Q variant.

### 2.3. Biological Investigations and Correlation with Genotype

Baseline T and DHT concentrations were determined for all the patients (100%), three of which had their baseline T and DHT measured more than twice. All but 1 of the 19 patients (94.7%) were subjected to hCG stimulation to obtain stimulated T/DHT ratios. These T, DHT, and T/DHT values at baseline and after hCG stimulation are presented in Appendix A. The median T/DHT ratios at baseline and after hCG stimulation were 5.2 and 17.4, respectively. As baseline T levels were undetectable at the prepubertal age, the minimum T/DHT ratio at baseline was <0.5. Conversely, after hCG stimulation, the T/DHT ratio of all the participants was >7.0. 

The median T, DHT, and T/DHT ratio values before and after the stimulation test are also presented according to age in Appendix A. In addition, the median T/DHT ratios of patients (6 months–12 years) with or without the p.R227Q variant are presented in Figure 3. Although not statistically significant, the T and DHT levels after hCG stimulation in the p.R227Q variant group were higher than those in the non-p.R227Q group. In contrast, following hCG stimulation, the T/DHT ratio was lower in the p.R227Q group than in the non-p.R227Q group; however, this difference was not statistically significant (Figure 3b). During the mini-puberty (before 6 months) and puberty (over 12 years old) periods, the difference between the p.R227Q and non-p.R227Q groups was incomparable because all patients in the mini-puberty period were in the non-p.R227Q group, while all but two in the puberty period were in the p.R227Q group. 

### 2.4. Genotype-Phenotype Correlation According to Structural Categories

Table 2 shows the classification of variants in SRD5A2 according to the structural categories [21] and the EMS values of cases reported in past studies, according to the mutations of each amino acid. 

Catalytic site mutations:

The p.Q56R, p.E57Q, p.E57D, and p.Y91D mutations belonged to this category. Although there was only one case of each of these mutations in each of the previous studies, the EMS range was between 2.0 and 3.0, indicating a severe female phenotype. 

2.NADPH-binding residue mutations:

Eight variants, including the p.R227Q mutation, were classified in this category. The EMS range of the mutations was wide and ranged between 2.0 and 8.0. However, except for p.N193S and p.H231R, all the EMS values were more than 4.0, indicating a mild to moderate phenotype for this category. Moreover, the EMS of the p.N193S mutation increased to 7.0 when combined with p.R227Q.

3.Structure-destabilizing mutations:

Seven variants, including the p.R246Q mutation, were classified in this category. Although the EMS range was wide, that of the homozygous mutations was relatively narrow (2.0–4.2). Their EMSs were relatively low and indicated a severe feminized phenotype. Conversely, in the cases of compound heterozygous mutations, the EMSs increased to 5.0–8.0.

4.Helix-breaking mutations:

Four variants were classified in this category. Very few cases have been reported with this mutation category, and the EMSs were low (2.0–3.0). In addition, functional studies have reported nearly null enzymatic activities associated with the mutations in this category, indicating severe phenotypes.

5.Small to bulky residues:

Five variants, including p.G203S, were classified in this category. In homozygous mutations, the EMS range was 2.0–3.3, indicating a severe feminized phenotype. However, the EMSs increased to 4.0–6.0 in compound heterozygous mutations, indicating a moderate phenotype.

### 2.5. Crystal Structure of the Human SRD5A2 Gene and Changes in Structure Relative to Variants

The crystal structure of the human wild-type SRD5A2 (SRD5A2-WT) in complex with NADPH is presented in Figure 4a. According to the 3D model of SRD5A2-WT and the NADPH-binding complex, when the 227th amino acid in the *SRD5A2* gene is arginine, the intra-atom distance between the ribose ring of NADPH and p.R227 residue of SRD5A2-WT is 8.4 Å (Figure 4b). Conversely, the substitution of this amino acid with glutamine resulted in this distance increasing to 9.6 Å (Figure 4c). This suggested that the essential interaction between SRD5A2 and NADPH changed as a result of the variant. Furthermore, the crystal structure showed that p.R246 was located in the cytosolic loop, and it is known to form multiple hydrogen bonds with residues in L5 [20]. The p.R246Q variant altered the residues, which may have led to the breakage or loosening of the bonds. (Figure 4d,e).

## 3. Discussion

To date, no genotype-phenotype correlation has been identified in 5αRD2. In this study, phenotype severity was compared according to the presence or absence of the p.R227Q and p.R246Q variants, which are most commonly found in the Korean population. Phenotype severity was also evaluated relative to the category of the SRD5A2 crystal structure. The patients with the p.R227Q variant had higher EMSs and lower incidences of hypospadias and cryptorchidism than those without the p.R227Q variant. This indicated that the p.R227Q variant exhibits a milder phenotype. Moreover, this variant was categorized as an NADPH-binding residue mutation, and nearly all the variants in this category exhibited a mild to moderate phenotype. On the contrary, the p.R246Q variant exhibited a more severe phenotype and was categorized as a structure-destabilizing mutation. Additionally, all the variants categorized as small to bulky residue mutations presented with moderate to severe phenotypes. Furthermore, the variants categorized as catalytic site and helix-breaking mutations exhibited severe phenotypes. To our knowledge, this is the first study to evaluate the correlation between phenotype severity and genotype from the perspective of these SRD5A2 structural categories.

The *SRD5A2* gene is located on chromosome 2p23 [26] and consists of five exons and four introns. Allelic variants have been reported in the entire gene, and loss of function related to either homozygous or compound heterozygous mutations has been reported. Currently, it is well established that the proportion of homozygous mutations is higher (60–70%) than that of heterozygous mutations in 5αRD2 [10,12,14,19,36]; however, an opposite ratio of homozygous (31.6%) and compound heterozygous mutations (68.4%) were identified in the present study. This is consistent with several recent East Asian cohort studies wherein the frequency of compound heterozygosity (55–80%) was higher than that of homozygosity [7,8,11,13,19]. In addition, although the mutations were distributed along all exons, exons 1 and 4 were considered hot spots in *SRD5A2* gene mutations [8,10,11,12,13,14,15]. However, in Turkish population studies [15,37], exon 3 has also been identified as a hot spot. In the present study, exon 5 as well as exons 1 and 4 were hot spots, which was consistent with a large cohort study from China [11]. The distribution of common variants in the *SRD5A2* gene has also been found to vary across ethnic groups. Positions 196, 227, 235, and 246 are hotspots of the *SRD5A2* gene worldwide [12]. However, p.R227Q along with p.Q6* are considered founder mutations in East Asia, and these two mutations account for 50–60% of the total mutations of the gene [8,11,13,14,22,38]. Conversely, p.G196S has been suggested to be the founder mutation in Turkey [15]. Therefore, genetic distributions in 5αRD2, including variant zygosity, hot spots, and common variants, not only vary geographically but also ethnically. Moreover, the phenotypic severity has been found to be relatively milder in Asian populations than in the Turkish and Italian populations [15]. A study on the Italian population found that more than half of the patients with 5αRD2 exhibited predominantly female phenotypes and sex assignments [39]. Studies on the Turkish population have further reported that almost half of the patients in the studies were initially reared as female and assigned to the female sex comprised one-third of the population [15]. On the contrary, in studies on Chinese cohorts, female external genitalia were found in only 10–20% of the patients with 5αRD2 [7,13]. This suggests that geographic and ethnic genetic differences may lead to variations in 5αRD2 phenotypic severity [15]. For example, with p.R227Q being a founder mutation in Asian populations, and p.G196S a founder mutation in Turkey [8,11,13,14,15,22,38]; these differences in founder mutations may lead to differences in the severities of phenotypes. Furthermore, this inconsistency suggests that there may be a correlation between genotype and phenotype.

To evaluate the genotype-phenotype correlation in 5αRD2, the molecular mechanisms and associated pathological effects of variants should be considered. Recently, a study on the crystal structure of human SRD5A2 [20] facilitated an evaluation of the molecular mechanisms underlying its enzyme activity. Human SRD5A2 consists of seven transmembrane α-helices (7-TMs) that are connected by six loops (L1-6) [20]. There is a large substrate-binding cavity inside the 7-TM domain, formed by all 7-TMs and L1, L3, and L5, with only one opening between TM1 and TM4 on the side of the 7-TM domain [20]. The cavity shows two relatively separate tunnel-like pockets for cofactor and substrate binding [20]. During enzymatic reactions, T accesses the ligand-binding pocket through the opening between TM1 and TM4, and the NADPH is close to the ∆4,5 bond of T [20]. The p.E57 in TM2 and p.Y91 in TM3 form hydrogen bonds with the C-3 carbonyl group of T, which is subsequently polarized to facilitate hydride transfer from NADPH to T, leading to a reduction of the ∆4,5 bond in T [20]. The binding pocket for NADPH is enclosed by the cytosolic loops, L1, L3, and L5, which are presumed to be physical barriers for NADPH/NADP^+^ exchanges. L1 undergoes dramatic conformational changes during the reaction to allow NADPH/NADP^+^ exchange; thus, L1 may be a “gate” domain that controls NADPH/NADP^+^ exchange [20]. Considering this molecular mechanism, p.E57 and p.Y91 may be categorized as catalytic site abnormalities [21], and mutations p.E57Q and p.Y91D can reduce enzymatic activity [20,26]. In contrast, p.R227 is located on TM7, and p.R227Q disrupts the hydrogen bonds with NADP^+^ [20]; therefore, such mutations may be categorized as NADPH-binding residue mutations [21]. Furthermore, p.R246 is located in the C-loop and forms multiple hydrogen bonds with the residues in L5 [20]; therefore, p.R246Q destabilizes L5 [20] and may be categorized as a structure-destabilizing mutation [21]. Moreover, G203 is located on the TM6 interface, and therefore the p.G203S mutation belongs to the small to bulky residue (destabilizing mutations) category [21].

In the present study, no variant belonging to the catalytic site and helix-breaking mutation category was identified. Nevertheless, four variants belong to the catalytic site mutation category [21], and, among them, the pathogenic mechanisms of p.E57 and p.Y91 have been identified through crystal structures [20]. Although there has been a limited number of cases presenting mutations of these sites, their phenotypes are close to the female phenotype with low EMSs [8,26,27,28]. Similarly, variants belonging to the helix-breaking category exhibit severe phenotypes. Only four variants are known to affect helix breaking [21], and a limited number of cases with these mutations have been reported [10,30]. However, in mutagenesis studies, enzymatic activity was nearly diminished [25,26,29], and the phenotypes presented as severe [10,27,30]. Therefore, it is suggested that the mutations affecting catalytic sites and helix breaking may be rare and associated with severe phenotypes.

Conversely, the phenotype of most variants belonging to the NADPH-binding residue category presented with mild to moderate severity. The p.R227Q variant, which was the second most frequent mutation in this study, belonged to this category. The p.R227 mutation is located in TM7, and the p.R227Q variant is known to significantly reduce SRD5A2 activity by affecting cofactor-binding affinity [19,20]. In the p.R227Q variant, the distance between the residue and NADPH increased from 8.4 to 9.6 Å (Figure 4), thereby disrupting hydrogen bond formation with NADP. In the present study, the EMSs of the p.R227Q variant group were higher than those of the non-p.R227Q variant group. In addition, the incidences of severe hypospadias and cryptorchidism in the p.R227Q variant group were lower than those in the non-p.R227Q variant group. Notably, all the patients with isolated micropenis had the p.R227Q variant, and all the female sex-assigned patients belonged to the non-p.R227Q variant group. The characteristically milder phenotype of the p.R227Q group compared to that of the non-p.R227Q group was consistent with that found in previous studies [8,11,13]. Furthermore, the milder phenotypes associated with Asian 5αRD2 patients compared to those associated with Europeans [15] may be ascribed to the fact that p.R227Q is a founder mutation in Asian countries [7,8,13,38]. Intriguingly, almost all the variants combined with p.R227Q presented mild to moderate phenotypes (Table 2), which suggested that compound heterozygous mutations with p.R227Q mitigate severe phenotypes. Several variants belonging to this category, such as p.N160D, p.R171S, and p.Y235F, also presented with a relatively moderate phenotype.

Seven mutations in the structure destabilization category were associated with NADPH and/or T binding abnormalities, shortened protein half-lives, and altered optimum pH. Among these mutations, p.R246Q is highly prevalent in a wide variety of geographical and ethnic backgrounds [19,23,34,40]. In the present study, p.R246Q was the most frequently occurring mutation, accounting for nearly half (17/38, 44.7%) of the alleles. Additionally, the R246Q/W mutation decreased NADPH-binding affinity and impaired SRD5A2 activity [20,25,34,41,42]. As the residual activity varied [25,26,34], the degree of external genital masculinization also varied [19]. However, homozygous p.R246Q/W showed a generally moderate to severe phenotype, having EMSs between 2.0–3.7 (Table 2) [8,10,13,15]. Conversely, the compound heterozygous mutation combined with another allele exhibited a milder phenotype than the homozygous mutation did [8,13]. In large Chinese cohorts, the median EMS of homozygous p.R246Q was found to be 3.0, while that of the compound heterozygous p.R246Q mutation combined with another allele was 3.0–7.0 [8,13]. Variants in the same category also presented moderate to severe phenotypes with an EMS of 2.0–4.2 in homozygous mutations (Table 2). The variants categorized as small to bulky residue mutations showed similar patterns. Among the variants in this category, the p.G203S mutation is frequently found in East Asian patients, including those from China [8,13,19,23,24]. In vitro functional assays showed that the enzymatic activity of this mutation decreased to approximately 60% compared to that of the wild-type [6]. Therefore, homozygous p.G203S exhibited marked feminization [6,8,14]. Homozygous mutations of the other variants belonging to the small to bulky residue category also showed severe phenotypes with EMSs of 2.0–3.3 (Table 2). On the contrary, patients with compound heterozygous variants belonging to this category reportedly presented with a more masculine phenotype [8,13,19,23,24,43]. Therefore, although it is too early to conclude, it is thought that homozygous mutations of these variants belonging to the structure-destabilizing and small to bulky residue categories exhibit severe phenotypes; however, the severities of these phenotypes may be mitigated by various alleles in compound heterozygous mutations.

It seems that there is a genotype-phenotype correlation pertaining to the structural category. Nevertheless, the severities of phenotypes may still vary within the same category. This is due to other factors, such as androgen receptor-mediated signal transduction activity, local concentrations of T in utero, the “backdoor” pathway of DHT, and environmental factors, all of which contribute to phenotypic variations [14,33]. Nonetheless, the structural categories of these variants are promising predictors of, at least, phenotypic severity.

### Limitations and Future Prospects

Given that 5αRD2 is a rare disorder, a limited number of participants were involved in this study. This may have compromised the reliability of our genotype-phenotype correlation results. However, similar conclusions were obtained by careful summarizations and analyses of previously published results, which overcame this limitation and strengthened our results. Further large-cohort studies focusing on the structural categories will be helpful in determining the genotype-phenotype correlation in 5αRD2. Second, the varying phenotypic severity of the homozygous and compound heterozygous p.R227Q variant groups was not identified in this study since only one patient had the homozygous p.R227Q variant. Previous studies have reported that there are no significant phenotypic differences between homozygous and compound heterozygous mutations [8,12,13]. However, it is still too early to conclude that homozygous and compound heterozygous mutations exhibit similar severities; thus, large-scale research is needed. Finally, even for compound heterozygous mutations, it is necessary to compare and analyze the differences between combined mutations in the same category and in different categories.

## 4. Materials and Methods

### 4.1. Patients

In this study, all participants visited the DSD clinic at Severance Children’s Hospital over a period of 20 years (from July 2003 to June 2022). Those participants who identified with 46,XY DSD and consented to genetic testing were included in this study. Those patients who had previously been genetically diagnosed with diseases other than 5αRD2 and those who refused genetic testing were excluded (Figure 5), and a total of 143 patients with DSD were enrolled. Molecular genetic analysis strategies have varied with the development of sequencing technologies and research aims. Therefore, targeted direct sequencing of the *SRD5A2* gene was mainly used before 2015, during which time two participants were molecularly diagnosed with 5αRD2. Thereafter, in 2016, DSD gene panel-based next-generation sequencing (NGS) was introduced to this center. Among the 143 participants, 67 were subjected to targeted NGS focused on DSD-related gene regions in 2016–2018 and 2020–2022, 10 of which (14.9%) were diagnosed with 5αRD2. In 2019, whole-exome sequencing (WES) was performed as part of a study aimed at identifying the causative gene of DSD. Those participants who were subjected to targeted NGS and those who were previously genetically confirmed with 5αRD2 were not included for WES. Seventy-four patients participated in the previous study, of which seven (9.5%) were molecularly diagnosed with 5αRD2. Finally, a total of 19 patients (13.3%) of 143 participants were genetically confirmed to have mutations in the *SRD5A2* gene, all of whom were clinically reviewed to have 5αRD2.

This study was approved by the Institutional Review Board of Severance Hospital Clinical Trial Centre (subject no. 4-2022-1340). In the previous study (n = 74), which was aimed at identifying the genetic cause of 46,XY DSD by WES, informed consent was obtained from the participating children’s parents. The current retrospective study only analyzed the results obtained from the medical records of the clinic; therefore, the requirement for informed consent of the other participants was waived. Nevertheless, informed consent was obtained upon collection of genetic information at the clinic. The current study was conducted in compliance with the Declaration of Helsinki to protect participant rights and personal information.

### 4.2. Evaluation of Clinical Manifestation

The 19 participants with confirmed 5αRD2 had ambiguous genitalia or sex inconsistencies, the 46,XY karyotype, no abnormality in the sex-determining Y region, normal adrenal function at initial evaluation, and high T/DHT ratios at baseline or after hCG stimulation.

We investigated the clinical manifestations, including external genital phenotype, family history, sex hormones before and/or after hCG stimulation, sex rearing, and treatment strategies such as surgeries, DHT cream application, and estradiol supplementation. Upon their first visit, the external genital phenotypes of the participants were classified using EMS [44], which is a sum of the external genitalia scores (i.e., presence of scrotal fusion, micropenis, and location of urethral meatus and both gonads). The score ranges between 0 and 12, and those close to 12 are considered as normal male phenotypes. The phallus length was measured in an extended state from the pubic symphysis to the top of the glans along the dorsal side. Micropenis was defined as phallus length < −2.5 SD relative to the average phallus length of same age Korean males [45]. Cryptorchidism was confirmed via physical and ultrasound examinations and was classified as intra-abdominal or inguinal based on the location of the testes. Hypospadias was divided into perineal, scrotal, penoscrotal, proximal, midshaft, and distal hypospadias based on the position of the urethral opening. The serum T concentrations was measured using an electrochemiluminescence immunoassay (Cobas 8000 instrument, Roche Diagnostics, Mannheim, Germany). The serum DHT concentration was measured using liquid chromatography tandem mass spectrometry (Quest Diagnostics, Madison, NJ, USA). The hCG stimulation test protocol was as follows: the baseline T and DHT levels were measured at 9:00 a.m., and the IVF-C^TM^ (hCG, LG Chem Ltd., Seoul, Republic of Korea) was administered at 1000 IU/dose at the same time for 3 consecutive days (days 0, 1, and 2). On day 3, the stimulation T and DHT levels were measured at 9:00 a.m. All of the participants underwent the same assessment.

We also reviewed the genotype-phenotype correlation according to five crystal-structure categories of SRD5A2 [21]. Data obtained from previous studies of each mutation were selected from the HGMD database [9]. Since phenotype was classified using EMS, data that measured EMS and/or recorded clinical features in detail (to allow EMS calculations) were used.

### 4.3. Molecular Genetic Testing

We confirmed homozygous or compound heterozygous variations of the *SRD5A2* gene of 19 patients with 5αRD2 by conducting direct sequencing (N = 2), targeted panel sequencing (N = 10), and WES (N = 7).

Genomic DNA was extracted from peripheral blood leukocyte samples using the QIAsymphony DNA Midi Kit (Qiagen, Hilden, Germany). Subsequent genomic DNA quantification was performed using a Qubit BR dsDNA kit (Invitrogen, Carlsbad, CA, USA).

For direct sequencing, exons 1–5 of the *SRD5A2* gene were amplified by PCR and subsequently purified using a Cosmo PCR purification kit (Cosmo Genetech, Seoul, Republic of Korea). Using the same sequencing primer employed during PCR, the amplified products were sequenced using a DNA sequencer (Life Technologies, Carlsbad, CA, USA) and a Life Technologies PRISM dye terminator cycle sequencing reaction kit (PerkinElmer, Waltham, MA, USA). The reference sequence of the *SRD5A2* gene was based on GRCh37/hg19 (GenBank accession number: NM_000348.3).

A customized gene panel, including 404 DSD-related genes (Appendix A), was designed for targeted sequencing. Target enrichment was performed using custom-designed RNA oligonucleotide probes and a target enrichment kit (Celemics, Seoul, Republic of Korea). Pooled libraries were sequenced using a NextSeq 550 sequencer (Illumina, San Diego, CA, USA) and a NextSeq Reagent Kit, version 2 (300 cycles).

For WES, the NGS library was prepared in accordance with the published commercial protocol of the Human Core Exome kit (Twist Bioscience, San Francisco, CA, USA). Over 80% of human exomes are less than 200 bases in length; therefore, we ran 70–100 million paired-end reads per sample. The blood-derived DNA libraries were sequenced using a Nova 6000 sequencer (Illumina, San Diego, CA, USA), achieving approximately 170 million reads per sample. Sequencing with a 151-bp, dual-indexed, paired-end sequencing configuration was performed. The bioinformatics pipeline, alignment processes, and quality procedures have been described elsewhere.

For each targeted NGS and WES sample, the raw sequencing data were saved in FASTQ format. Phred quality score was used to determine the read accuracy of the reference genome sequence [46]. The raw sequencing data were trimmed and filtered using Trimmomatic software. Sequences were aligned to the hg19 reference genome using Burrows-Wheeler Aligner (BWA-aln) and Bowtie [47,48]. Post-alignment processing was performed on sequence alignment/map (SAM) files using SAMtools [49], whereafter the alignments were converted to the binary alignment/map (BAM) format. Single nucleotide variants and small insertions or deletions were called and crosschecked using GATK software version 3.8.0 with Haplotypecaller [50] and VarScan version 2.4.0.

The called variants were annotated mutation records and population frequencies obtained from public databases, such as HGMD [9], ClinVar [51], Exome Sequencing Project [52], 1000 Genomes Project [53], Genome Aggregation Database (gnomAD) [54], and Korean Reference Genome database [55]. Thereafter, deleterious effects of each annotated variant were predicted using 5 in silico prediction algorithms, including SIFT, PolyPhen2, FATHMM, and CADD [56].

Those variants that adhered to the following criteria were selected: (1) an allele frequency < 1% in 1000 Genomes Project and gnomAD; (2) not available in our in-house database; (3) protein-altering variants; and (4) a high read quality (defined as read number > 20 or quality score [QS] > 30). Candidate variants were further confirmed using Sanger sequencing as described above. If the variants were suspected compound heterozygotes, the parents of the participants were also examined to determine the inheritance pattern. Finally, the candidate variants were interpreted in accordance with the 5-tier classification system recommended by the American College of Medical Genetics and Genomics and the Association for Molecular Pathology [57]. The pathogenicity of those variants identified in this study were determined as per the rules for combining criteria to classify variants: ‘pathogenic’ and ‘likely pathogenic’ were selected.

Participants with 5αRD2 were classified as either p.R227Q variant (N = 9) or non-p.R227Q variant (N = 10) and as either p.R246Q variant (N = 13) or non-p.R246Q variant (N = 6) to compare whether a difference in clinical characteristics existed between these groups.

### 4.4. 3D Protein Structure of SRD5A2 with NADPH and Computational Mutagenesis

The crystal structure of human *SRD5A2* in complex with NADPH was retrieved from the protein data bank (PDB; PDB ID: 7BW1). Computational mutagenesis and measurements of intra-atom distance between the variant residues and ribose ring of NADPH were performed using PyMOL (Version 2.5.4; Schrödinger Inc., New York, NY, USA; https://pymol.org/2/ accessed on 1 January 2023).

### 4.5. Statistics Analysis

Statistical analyses were performed using SPSS (Version 26.0; IBM Corp., Armonk, NY, USA). The Mann-Whitney *U* test was used for nonparametric data analysis and is presented as median values and as ranges (minimum and maximum). Fisher’s exact test was used to compare the incidence between sex rearing and clinical manifestations. Statistical significance was based on *p* < 0.05.

## 5. Conclusions

Although many attempts have been made to evaluate the genotype-phenotype correlation in 5αRD2, such a correlation is yet to be demonstrated. In this study, the p.R227Q variant presented a more masculine phenotype and mitigated the phenotypic severity. Other mutations in the NADPH-binding residue mutation category, to which the p.R227Q variant belonged, also showed a mild to moderate phenotype. In contrast, variants of the structure-destabilizing mutations and small to bulky residue categories exhibited a moderate to severe phenotype, while the variants of catalytic site and helix breaking mutations presented with severe phenotypes. The SRD5A2 structural approach utilized in this study suggested the existence of a genotype-phenotype correlation in 5αRD2. To the best of our knowledge, this is the first study to evaluate the genotype-phenotype correlation from a structural perspective of SRD5A2. This genotype-phenotype correlation is of great value to the clinical management and treatment of 5αRD2, including sex assignment and genetic counselling. Therefore, further studies using the structural perspective of SRD5A2 are necessary to improve clinical management of 5αRD2.

## Figures and Tables

**Figure 1 ijms-24-03297-f001:**
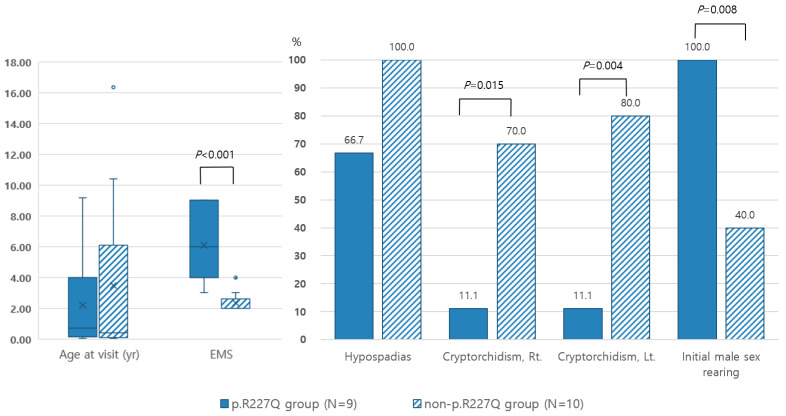
Clinical manifestations of 19 patients with or without the p.R227Q allele. × and horizontal line represented the mean and median values, respectively. ● expressed outlier data.

**Figure 2 ijms-24-03297-f002:**
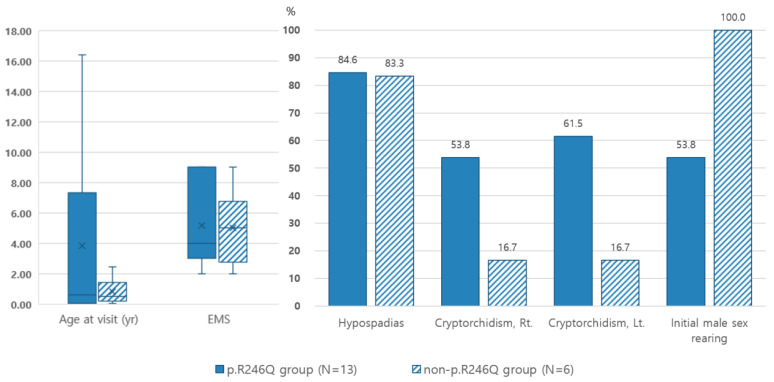
Clinical manifestations of 19 patients with or without the p.R246Q allele. × and horizontal line represented the mean and median values, respectively.

**Figure 3 ijms-24-03297-f003:**
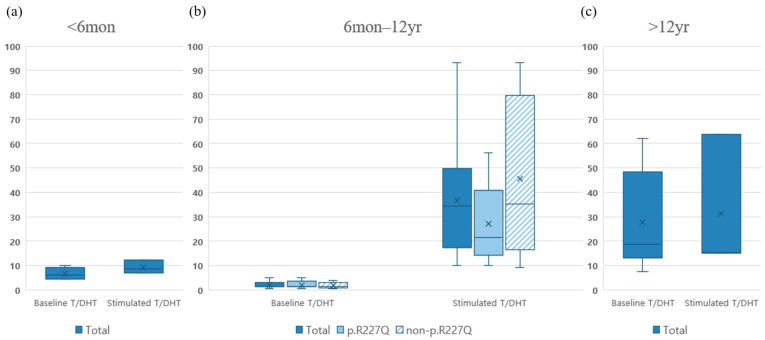
Median T/DHT ratios at baseline and after hCG stimulation, grouped according to age and with or without the p.R227Q allele. (**a**) <6 months old (N = 4 at baseline and N = 3 after stimulation, all patients were in the non-p.R227Q variant allele group). (**b**) 6 months–12 years old (N = 12 at baseline [N = 7, p.R227Q; N = 5, non-p.R227Q] and N = 13 after stimulation [N = 8, p.R227Q; N = 5, non-p.R227Q]). (**c**) >12 years old (N = 11 at baseline and N = 3 after stimulation, only two participants were in the non-p.R227Q variant group). × and horizontal line represented the mean and median values, respectively.

**Figure 4 ijms-24-03297-f004:**
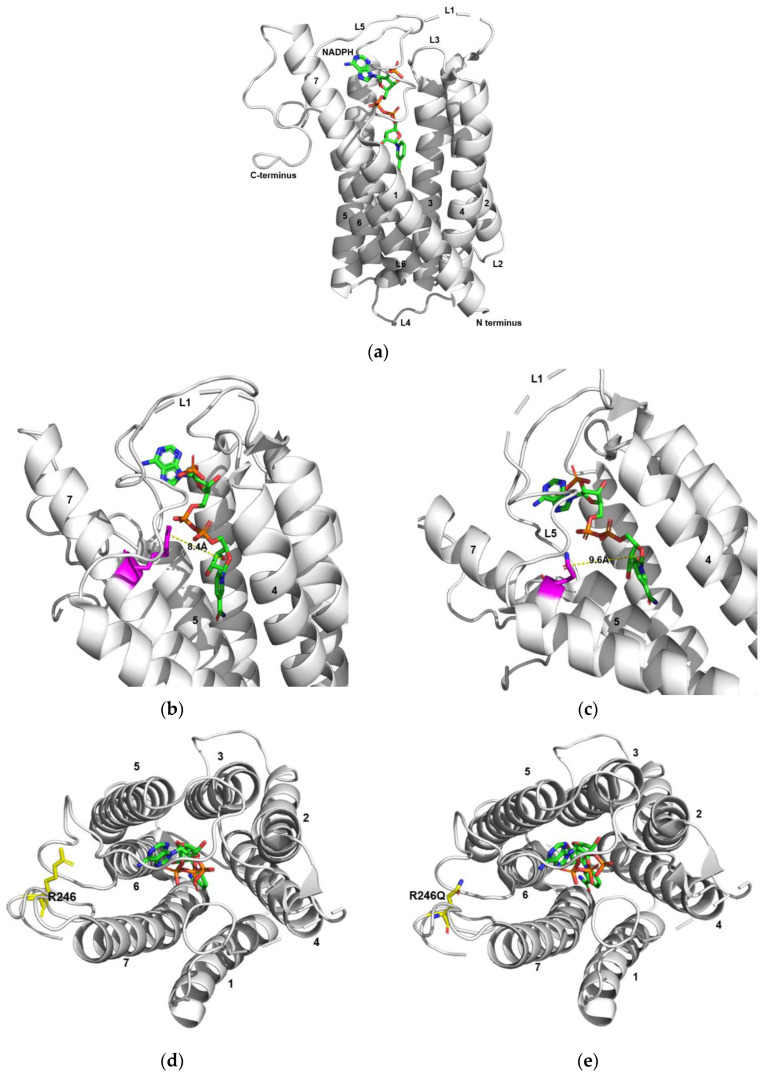
Overall structures of human SRD5A2 and NADPH. (**a**) Vertical perpendicular view of the SRD5A2-NADPH complex structure. (**b**) The distance from NADPH when p.227 is arginine (R) (wild-type). (**c**) Distance from NADPH when p.227 is glutamine (Q). (**d**) Horizontal perpendicular view of the SRD5A2 structure with the wild-type p.246 and NADPH. (**e**) Horizontal perpendicular view of the SRD5A2 structure with the p.R246Q variant and NADPH from the top.

**Figure 5 ijms-24-03297-f005:**
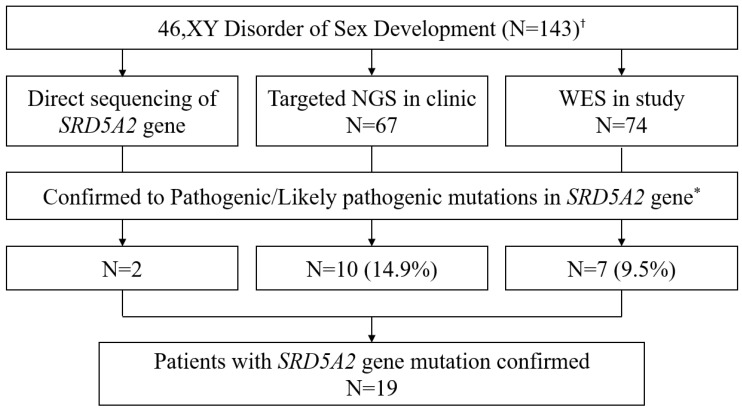
Participant selection for this study. ^†^ Patients who had already been diagnosed genetically with different diseases or refused genetic testing were excluded. * According to the American College of Medical Genetics and Genomics guidelines.

**Table 1 ijms-24-03297-t001:** Molecular and clinical data of patients with 5α-reductase deficiency.

Patient No.	*SRD5A2* Mutation	Exon	Sex of Rearing	Age at First Visit	Phenotypes at First Visit					EMS
					C/P	Hypospadias	Rt. Testis	Lt. Testis	P-S Disposition	
1	R227Q/R227Q	4/4	M	2.4Y	MP	Midshaft	Scrotum	Scrotum	Y	4
2	R227Q/T53R	4/1	M	1.1Y	MP	Perineal	Scrotum	Scrotum	Y	3
3	R227Q/A65P	4/1	M	4M	MP	Penoscrotal	Scrotum	Scrotum	N	6
4	R227Q/Q6*	4/1	M	21D	MP	Penoscrotal	Scrotum	Scrotum	N	6
5			M	9M	MP	None	Scrotum	Scrotum	N	9
6	R227Q/R246Q	4/5	M	5.5Y	MP	None	Scrotum	Scrotum	N	9
7			M	9.1Y	MP	None	Scrotum	Scrotum	N	9
8			M	10D	MP	Perineal	Inguinal	Inguinal	Y	5
9			M	5M	MP	Proximal	Scrotum	Scrotum	Y	3
10	R246Q/R246Q	5/5	M	10D	CM	Perineal	Scrotum	Scrotum	Y	3
11			M	8M	CM	Proximal	Scrotum	Scrotum	Y	3
12			M	7D	CM	Scrotal	Inguinal	Inguinal	Y	2
13			F	4.6Y	CM	Perineal	Inguinal	Inguinal	Y	2
14	R246Q/R246W	5/5	F->M	16.4Y	MP	Perineal	Inguinal	Inguinal	Y	2
15	R246Q/G203S	5/4	F->M	2M	CM	Perineal	Inguinal	Inguinal	Y	2
16			F	2M	C	Perineal	Scrotum	Inguinal	Y	2.5
17	R246Q/F219Sfs*60	5/4	F	2.5Y	C	Perineal	Inguinal	Inguinal	Y	2
18	R246Q/Q6*	5/1	F	10.4Y	C	Perineal	Inguinal	Inguinal	Y	2
19	Q6*/Q6*	1/1	M	3M	C	Scrotal	Abd	Abd	Y	2

C, clitoris; CM, clitoromegaly; EMS, external masculinization score; MP, micropenis; P, penis; P-S disposition, penoscrotal disposition.

**Table 2 ijms-24-03297-t002:** Classification of variants of *SRD5A2* according to various structural categories.

Structural Category [21]	Mutations in *SRD5A2*	Location [20]	Enzymatic Activity	Mechanism	EMS
Catalytic site mutations	p.Q56R	TM2	No [25,26] ^† (L55Q)^		*3.0 [27] ^‡^* *2.5 [26] ^† (L55Q)^*
	p.E57Q	TM2	Reduced [28]		*3.0 [28] ^† (G85D)^*
	p.E57D	TM2			*2.0 [8] ^† (R227Q)^*
	p.Y91D	TM3	No [25]	Shortening protein half-life [25]	*2.0 [26] ^† (D164V)^*
NADPH binding residue mutations	p.N160D	TM5			4.8 ± 3.0 [10] ^‡^
	p.D164V	TM5	No [25]		
	p.R171S	TM5	Reduced [25,26] ^† (G34R)^	NADPH-binding abnormality [25] Shortening protein half-life [25] Change optimum pH [25]	*5.0 [29] ^† (G196V)^* *5.0 [30] ^† (G196V)^*
	p.N193S	TM6	Reduced [25]	NADPH-binding abnormality [25] Shortening protein half-life [25] Change optimum pH [25]	*2.0 [27] ^‡^* 2.0 [13] ^†^ 7.0 [8] ^† (R227Q)^
	p.E197D	TM6	No [25,31]	Shortening protein half-life [25]	
	p.R227Q	TM7	Reduced [32]	NADPH-binding abnormality [32] Testosterone-binding abnormality [32]	8.0 ± 1.8 [10] ^‡^ 6.0 [13] ^‡^ 7.0 [8] ^‡^ 7.8 [8] ^† (G203S)^ 7.0 [8] ^† (R246Q)^ 7.0 [8] ^† (N193S)^ 4.0 [8] ^† (G34R)^ 6.0 [13] ^†^ 6.0 [8] ^†^
	p.H231R	TM7	Reduced [25]	Testosterone-binding abnormality [25] Change optimum pH [25]	2.0 ± 1.2 [10] ^‡^
	p.Y235F	TM7			4.0 ± 3.5 [10] ^‡^
Structure destabilizing mutations	p.Q126R	TM4	No [25,26]	Shortening protein half-life [25]	4.2 ± 1.5 [10] ^‡^
	p.P181L	TM5-TM6	Reduced [25]	NADPH-binding abnormality [25] Shortening protein half-life [25] Change optimum pH [25]	2.0 [16] ^‡^ 4.0 ± 2.8 [10] ^‡^ *6.0 [13] ^†^*
	p.G183S	TM5-TM6	Reduced [25]	Decrease of testosterone affinity NADPH-binding abnormality [25]	4.2 ± 2.5 [10] ^‡^ *8.0 [13] ^†^*
	p.A207D	TM6	No [25] Reduced [26] ^† (R246Q)^	Shortening protein half-life [25]	
	p.P212R	TM6	No [6,19,25,29,31,33]	Shortening protein half-life [25]	
	p.R246Q	TM7	Reduced [25,26]	NADPH-binding abnormality [25] Shortening protein half-life [25] Change optimum pH [25]	2.0 [15] ^‡^ 3.7 ± 1.7 [10] ^‡^ 3.0 [8] ^‡^ *2.5 [30] ^‡^* 5.0 [8] ^† (G203S)^ 5.0 [13] ^†^
	p.R246W	TM7	No [26] Reduced [25,34,35]	NADPH-binding abnormality [25,34] Shortening protein half-life [25] Change optimum pH [25]	2.7 ± 1.2 [10] ^‡^ 2.0 [13] ^†^
Helix breaking mutations	p.L55Q	TM2	No [25,26] ^† (Q56R)^	Shortening protein half-life [25]	3.0 ± 3.0 [10] ^‡^ *2.0 [27] ^‡^*
	p.H162P	TM5	Reduced [29]		2.0 [30] ^† (Q6*)^
	p.L224P	TM7	No [25,26]	Shortening protein half-life [25]	
	p.H230P	TM7	No [25]	Shortening protein half-life [25]	
Small to bulky residues	p.G34R	TM1-TM2	No [26] ^‡ † (G115D)^ Reduced [25,26] ^† (R171S)^	NADPH-binding abnormality [26] ^‡^ Testosterone-binding abnormality [25,26] ^† (G115D)^ Change optimum pH [25,26]	3.3 ± 2.1 [10] ^‡^ 4.0 [8] ^† (R227Q)^
	p.P59R	MT2	No [25]	Shortening protein half-life [25]	
	p.G115D	TM4	No [25,26] ^† (G34R)^	Testosterone-binding abnormality [26] ^† (G34R)^ (mild) Shortening protein half-life [25]	2.6 ± 0.6 [10] ^‡^
	p.G196S	TM6	Reduced [25,26]	NADPH-binding abnormality [25,26] Shortening protein half-life [25]	3.3 ± 1.0 [10] ^‡^ 5.8 [13] ^†^
	p.G203S	TM6	Reduced [6]		2.5 [8] ^‡^ *2.0 [29] ^‡^* 6.0 [13] ^†^ 5.0 [8] ^† (R246Q)^

The EMS is displayed in italics if only one case was reported. ^‡^ Homozygous mutation data. ^†^ Compound heterozygous mutation data.

## Data Availability

Not applicable.

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
