# Peer review of "The Genotype-Phenotype Correlation in Human 5α-Reductase Type 2 Deficiency: Classified and Analyzed from a SRD5A2 Structural Perspective"

_ijms, 2023, doi:10.3390/ijms24043297_

Round 1

Reviewer 1 Report

SRD5A2 (type II steroid 5α-reductase) plays a critical role in androgen metabolism and sex development. SRD5A2 convert testosterone (T) to 5α-dihydrotestosterone (DHT), the more potent androgen required to develop external male genitalia. SRD5A2 deficiency is related to disorders of sex development. 

This study evaluated the genotype-phenotype correlation from a structural perspective in 19 Korean patients with type II 5α-reductase deficiency. This study compared clinical findings between patients with or without p.R227Q and p.R246Q variants, which were the most prevalent in these Korean patients. In addition, they evaluated the correlation between genotype and phenotype by classifying data from previous studies reported in the HGMD database according to the crystal structure of SRD5A2 and then comparing the severity of phenotypes. From the perspective of structural biology, the authors divided the mutation types into five categories based on the position of mutation sites on proteins and their effects: (1) catalytic site mutations, (2) NADPH-binding residue mutations, (3) structure destabilizing mutations, (4) helix breaking mutations, and (5) small to bulky residue (destabilizing mutations). This study suggests that catalytic site and helix-breaking mutations exhibited severe phenotypes. In addition, small to bulky residue mutations also presented moderate to severe phenotypes. The p.R227Q variant, which belongs to the NADPH binding residue mutation category, exhibits a milder phenotype. The p.R246Q variant, categorized as a destabilizing structure mutation, showed a more severe phenotype.

This paper is the first study to analyze the correlation between the genotype and phenotype of type II 5α-reductase deficiency from a structural perspective. It provides a new reference for the diagnosis and treatment of this disease. However, the authors must conclude carefully due to the small sample size. This work could be accepted after minor revisions. Major concerns are shown below. 

1. The small sample size of 19 patients is insufficient to support the current conclusion. The author should add more descriptions of the previous study to support the current conclusion. 

2. The effects of different ethnic and geographical differences on the same mutation mentioned in the paper are not given specific data.

3. The authors need to give a general introduction of the specific course of hCG treatment in the main text for a better understanding of who is outside this field. In addition, detailed parameters in the method should also be described, such as treatment time and dosage.

4. There is no description of what samples were collected from patients to measure the concentrations of T, DHT, and the T/DHT ratios in this article.

5. Due to the small sample size, the comparison results on the severity of phenotypes caused by homozygous and heterozygous mutations in this paper are insufficient to support the conclusion.

6. The author showed the p.R246Q variant altered the structure (Figure 4d-e). However, specific structural changes have not been described. In addition, the method has no description for predicting structural changes of R246Q mutants.

Reviewer 2 Report

The manuscript “The genotype-phenotype correlation in human 5α-reductase type 2  deficiency: classified and analysed from a SRD5A2 structural perspective” is a very well designed and presented study based on 19 Korean patients carriers of biallelic SRD5A2 gene mutations with variable phenotypes of 5α-reductase type 2  deficiency. They also reviewed literature to discuss published data on genotype-phenotype associations in other series.

Authors classify mutations according to structural categories that condition functional effects. Thereafter, they analyse the different phenotypic characteristics as well as the hormonal phenotypes of their patients, as classified according to their mutational structural categories. This afforded a classification into degrees of enzyme deficiency that correlate with several phenotypic characteristics.

This procedure would allow a finer prediction of severity of enzyme deficiency which is relevant for the management and genetic counselling of this category of affected patients.

Manuscript is very well presented.

Only minor changes:

- 1) Abstract, lines 19-20: I would not be so radical “has not yet still been evaluated”. I would say “has not yet still been adequately evaluated”

- 2) Abstract, line 21: “discovered”, wouldn’t it be “described” ??

- 3) Introduction, line 39: introduce “disorders/difference of sex development (DSD), …”

- 4) Introduction, line 53: change “determination” by “analysis”.

- 5) Results, line 92: “From 143 patients with 46,XY DSD, ….”

- 6) Figure 2: indicate the statistical significance, as on Fig. 1.

- 7) Figure 3: indicate a, b and c on the Figure.

- 8) Results, lines 193 and 290: “nearly diminished” should be “nearly null”.
